# Investigating the Linear Dynamics of the Near-Field of a Turbulent High-Speed Jet Using Dual-Particle Image Velocimetry (PIV) and Dynamic Mode Decomposition (DMD)

Vishal Chaugule [1], Alexis Duddridge [1], Tushar Sikroria [1,2], Callum Atkinson [1] and Julio Soria [1,*]

1   Laboratory for Turbulence Research in Aerospace and Combustion (LTRAC), Department of Mechanical and Aerospace Engineering, Monash University, Clayton, VIC 3800, Australia
2   Von Karman Institute, 1640 Sint-Genesius-Rode, Belgium
*   Correspondence: julio.soria@monash.edu

**Abstract:** The quest for the physical mechanisms underlying turbulent high-speed jet flows is underpinned by the extraction of spatio-temporal coherent structures from their flow fields. Experimental measurements to enable data decomposition need to comprise time-resolved velocity fields with a high-spatial resolution—qualities which current particle image velocimetry hardware are incapable of providing. This paper demonstrates a novel approach that addresses this challenge through the implementation of an experimental high-spatial resolution dual-particle image velocimetry methodology coupled with dynamic mode decomposition. This new approach is exemplified by its application in studying the dynamics of the near-field region of a turbulent high-speed jet, enabling the spatio-temporal structure to be investigated by the identification of the spatial structure of the dominant dynamic modes and their temporal dynamics. The spatial amplification of these modes is compared with that predicted by classical linear stability theory, showing close agreement, which demonstrates the powerful capability of this technique to identify the dominant frequencies and their associated spatial structures in high-speed turbulent flows.

**Keywords:** turbulent high-speed jet; particle image velocimetry; dynamic mode decomposition

## 1. Introduction

Turbulent high-speed jets exist in diverse applications in fields ranging from aerospace [1] to pharmaceutics [2]. In addition, within fundamental fluid dynamics theory turbulent free jet flows are a canonical example of a turbulent free shear flow. The large-scale coherent structures in these turbulent jets play an important role in the transport and mixing of active and passive scalars [3–6], as well as aerodynamic noise generation [7–9]. Experimental studies of the interaction and evolution of these structures provides a direct representation of the flow's governing physics. However, performing a quantitative study of the flow's governing physics requires high-resolution data in both the spatial and temporal dimensions.

One diagnostic method that is now becoming a standard experimental fluid mechanics measurement tool is particle image velocimetry (PIV), which is a widely used whole-field technique that tracks the motion of a group of particles within a small flow measurement volume to determine instantaneous velocity vector fields [10–12]. PIV measurements of turbulent free-jets with high-spatial resolution have been possible due to the increased resolution (10–30 Mpx) and sensitivity of new imaging sensors [7,13–15], where the spatial resolution constitutes an important parameter that is necessary for the accurate evaluation of the turbulence statistics, velocity gradients and vorticity in turbulent shear flows with minimal filtering and low uncertainty [16–18]. Time-resolved PIV measurements, enabled by high-repetition rate lasers and high-speed cameras (1–25 kHz), have also been used to examine the development and evolution of large-scale structures in a turbulent free-jet shear

layer [19–23]. However, in both these types of PIV measurements, the resolution in the other dimension is limited by the current technology of image acquisition sensors, wherein the time resolution of the measured data needs to be sacrificed to obtain high-spatial resolution, and vice versa.

The identification of coherent and dominant flow features in turbulent free-jets provides valuable information to decipher the underlying flow dynamics. Proper orthogonal decomposition (POD) of the velocity field is commonly used to identify coherent flow structures based on a hierarchical ranking of their turbulent kinetic energies [24,25]. POD of turbulent free-jet velocity fields, obtained from PIV measurements, has been used in previous studies [26–32] to identify typical flow patterns or dominant modes from the seemingly incoherent velocity fluctuations of the jet. However, one of the drawbacks of POD is that the extracted energetically dominant modal structures will not necessarily be dynamically relevant in all circumstances, in addition to the loss of phase information since POD is based on the use of second-order correlations [33]. In contrast, dynamic mode decomposition (DMD) is a more recently introduced technique by Schmid [34] that identifies dynamic flow structures each with a single characteristic frequency of oscillation and its corresponding temporal growth/decay rate. DMD of PIV velocity fields of turbulent free-jets has been undertaken in only a few previous studies [35–37]. As the shear layers are remarkably thin $O(10^{-3})$ with respect to the jet-exit diameter in the near-nozzle region [38], high-spatial resolution of any time-resolved PIV measurements is essential in order to resolve and detect any coherent structures within the turbulent shear layer, which is a difficult task with the current high-speed image acquisition technology.

While we look forward to the next generation of imaging sensor technology to eventually overcome the limitation of simultaneous high-spatial resolution with high-speed image acquisition, it is prudent to leverage the current state of image acquisition technologies coupled with advanced data analytics to enable a comprehensive experimental spatio-temporal characterisation of turbulent high-speed free-jet flows, as well as other high-speed turbulent shear flows. The original DMD algorithm outlined by Schmid [34], which will be referred to as *Standard DMD*, derives the best linear approximation to the mapping matrix between a sequential set of time-resolved data vectors. This algorithm was subsequently generalised by Tu et al. [39], referred to as *Exact DMD*, for application to non-sequential time series that comprise time-resolved data vectors collected as a set of pairs. The acquisition of such pairs of flow-field data vectors, which for our requirement are time-resolved velocity fields at sufficiently high-spatial resolutions, can be realised by a *Dual-PIV* system. This type of *Dual-PIV* system using polarisation-based image separation has been used in previous studies [40–43], albeit using low-spatial resolution image sensors to determine space–time correlations and velocity accelerations. Basically, this system comprises two independent, but synchronised, PIV systems illuminating and recording the same field of view with cross-talk minimised using polarisation-based image separation. The timing interval between these two PIV systems can be made sufficiently small to resolve down to the smallest dynamically significant temporal scale.

The novelty of this study is the combination of *Dual-PIV* measurements, which constitute an ensemble of two time-resolved velocity fields at a high-spatial resolution, coupled with their *Exact DMD* analysis. The combined application of these methodologies is used to study the linear dynamics of a turbulent round high-speed free-jet. The aim is to extract the frequencies of interest and the corresponding spatial modes, which dictate the instabilities in the free-shear layer. The methodology presented is capable of a temporal resolution of 1 MHz at an image resolution of 15 Mpx, indicating a high spatio-temporal resolution of the flow-field measurements. The spatial growth rates of the energy of the DMD modes are computed to determine the most spatially amplified modes, i.e., the instabilities within the shear layer of a turbulent high-speed free-jet flow.

## 2. Experimental System

### 2.1. Jet Facility

The experiments were conducted in the sub/super-sonic jet facility at the Laboratory for Turbulence Research in Aerospace and Combustion (LTRAC), Monash University. This facility has been extensively used for fluid dynamics research in turbulent jet flows and its details can be found in refs. [15,44,45]. Figure 1 illustrates the schematic of this facility, wherein the jet emerges from a converging nozzle, with an exit inner diameter of $d = 15$ mm, into a full optical access enclosure made of Perspex walls. This enclosure has a square cross-section of length $42d$ and a height of $80d$ that terminates into a passive exhaust for the jet flow. The coordinates $x$ and $y$ represent the axial and lateral directions of the jet flow, respectively. The jet Reynolds number is $Re_d = 10{,}000$, where $Re_d$ is the Reynolds number based on the jet orifice diameter and the mean jet orifice velocity. An in-house developed Laskin nozzle particle generator that produces droplets of $\approx 1$ µm diameter by aerosolisation of paraffin oil is used to supply the tracer particles to seed the flow. These droplets have a diffraction limited minimum image diameter of 8 µm and a relaxation time of $\approx 4$ µs [46].

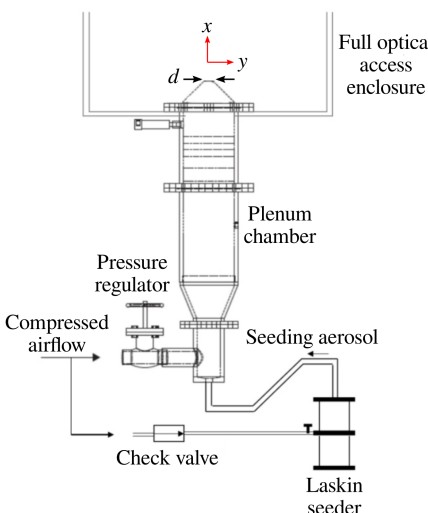

**Figure 1.** Schematic of the jet facility.

### 2.2. Dual-PIV

A schematic of the *Dual-PIV* setup is shown in Figure 2. A longitudinal plane in the seeded jet flow is illuminated using an *Innolas SpitLight DPSS EVO IV* laser for two-component–two-dimensional (2C–2D) PIV measurements. This is an optically and electronically integrated system of two orthogonally polarised PIV lasers in which each laser produces collinear pulsed-laser light. The polarization difference between the two lasers is leveraged to enable *Dual-PIV* measurements by utilizing twin cameras arranged around a polarizing beam-splitter. *Camera-1* records the single-exposed image pair of the flow-field illuminated only by the *MASTER* laser, while *Camera-2* records the single-exposed image pair illuminated only by the *SLAVE* laser. This optical imaging method has been previously used in experimental fluid mechanics [47–50]. Further details of the *Dual-PIV* laser are provided in ref. [51].

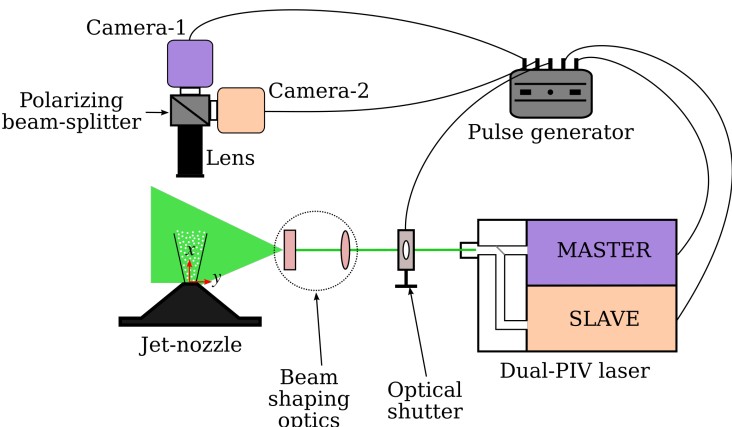

**Figure 2.** Schematic of the experimental setup.

The imaging system comprises twin *Imperx B4820 CCD* cameras (4872 px × 3248 px, 7.4 μm) and a single *Nikon AF Micro-NIKKOR 200mm f/4D IF-ED* lens mounted in front of the polarizing beam-splitter. The lens aperture and reproduction ratio were set to $f/4$ and 1:2.1, respectively, to image a field-of-view that spanned $\approx 5d \times 3.2d$ ($x \times y$). This region was located $0.17d$ downstream of the jet-nozzle exit to avoid laser-light reflections from the nozzle surface in the PIV images. The spatial resolution of the recorded images of both PIV image acquisition systems was $15.8 \pm 0.35$ μm/px at a PIV velocity field acquisition rate of $f_{acq} = 1$ Hz.

The minimum repetition rate of the *Dual-PIV* laser was 10 Hz, which required the use of an electro-mechanical beam shutter (*Thorlabs SH05 Shutter*) to generate a pulsed laser beam that was synchronous with the camera image acquisition rate. This shutter was placed between the laser and beam shaping optics, as shown in Figure 2, and was controlled with a *Thorlabs SC10 Controller*. The laser, the shutter and the twin cameras were synchronised with a *BeagleBone Black* (*BBB*)-based in-house-developed pulse generator [52] that has a deterministic temporal resolution of 5 ns.

A schematic of the timing configuration used for the *Dual-PIV* measurements is illustrated in Figure 3. The time between two pulses of the *MASTER* laser was set to $\delta t = 12$ μs, which was also used for the *SLAVE* laser. The time shift between the *MASTER* and *SLAVE* lasers was set to $\Delta t = 8$ μs, with pulse A of the *MASTER* laser taken as the reference. As the twin camera exposure times were synchronised to this shift, the time between the sequential velocity fields (snapshots) obtained from *Camera-1* and *Camera-2* was equal to $\Delta t$. Three such pairs of sequential snapshots are shown illustrated in the right side in Figure 3 with the velocity snapshots in the $k$th pair labelled as $\mathbf{v}_t^k$ and $\mathbf{v}_{t+\Delta t}^k$, respectively. The time between the velocity snapshot pairs is, however, equal to the reciprocal of the image acquisition rate ($1/f_{acq}$). The selected value of $\Delta t$ was based on the linear stability theory result that the most spatially amplified instability wave in a circular jet near-field shear layer has a Strouhal number $St_n = f_n\theta/U_j = 0.032$ [53]. For the present case, the momentum boundary layer thickness at the jet-nozzle exit is $\theta = 0.046d$ and the characteristic velocity is $U_j = 9.69$ m/s, which is the average velocity across the jet exit. The value of $\Delta t$ chosen was such that the natural frequency $f_n$ of the circular jet near-field shear layer was over-sampled at least 100 times the Nyquist rate ($2f_n$). A total of 10,000 velocity snapshot pairs were acquired.

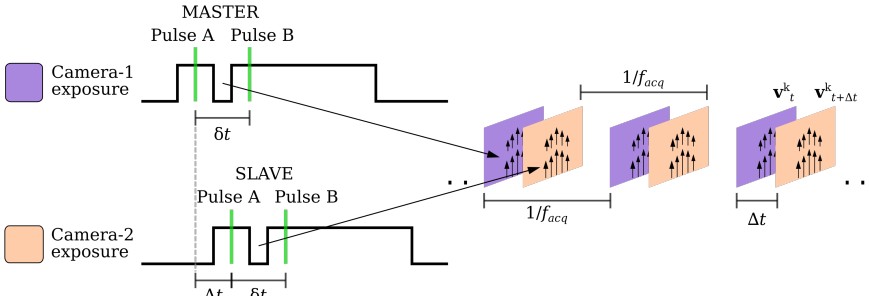

**Figure 3.** Dual-PIV timing diagram and pairwise time-resolved velocity field measurements.

### 2.3. PIV Processing Algorithm

Analysis of the PIV images was performed using multi-grid/multi-pass cross-correlation digital particle image velocimetry (MCCDPIV) introduced by Soria using a highly parallelised MPI/C++ program [54–56]. This PIV analysis method of single-exposed images employs an iterative and adaptive cross-correlation algorithm to increase the dynamic range of the measurements. This is achieved by adapting the sample window size to the local flow conditions and offsetting the discrete sampling window in the second frame by an amount approximately equal to the estimated particle displacement in the sampling window. For the present case, the sampling window sizes in a two-pass MCCDPIV analysis were (64 px × 16 px) and (32 px × 8 px) with a 0.5 overlap. The correlation-function-based validation of [57] was used to eliminate spurious vectors. A maximum displacement ratio of 0.3 in the final sampling window and a normalised median test [58] with a threshold value of 0.75 were applied to validate the resulting displacement vectors, which were also filtered to remove high-frequency noise using a Chi-Square velocity field fit over 9 data points [11,55,59]. The resulting grid spacing between the velocity vectors is $\approx 0.015d \times 0.004d$ ($x \times y$), resulting in a cross-stream velocity vector spacing of $O(0.1\theta)$.

### 2.4. Measurement Uncertainties

The uncertainty in measurement of the jet-nozzle exit inner-diameter is $\pm 0.079$ mm, and that of the flow rate into the jet nozzle is $\pm 3.82$ L/min$^{-1}$. The pulse generator has a root-mean-square jitter of 290 ps and a timebase stability on the order of 10 ppm [52], and the laser has a temporal jitter of $\pm 1$ ns. These give uncertainties of $\pm 1.044$ ns and $\pm 1.048$ ns for the time shift between the *MASTER* and *SLAVE* lasers ($\Delta t = 8$ μs) and for the time between two pulses of the *MASTER* (*SLAVE*) laser ($\delta t = 12$ μs), respectively. The MCCDPIV algorithm has a displacement measurement uncertainty of $\pm 0.06$ px at a 95% confidence interval for a sampling window size of 32 px [60] and the uncertainty in the optical magnification is $\pm 0.35$ μm/px. These uncertainties, together with that in $\delta t$, give an uncertainty of $\epsilon_{\rm u}/U_{\rm j} = 2.36\%$ in the PIV velocity measurement.

## 3. Dynamic Mode Decomposition

In order to find a low-dimensional approximation of the dynamics in the form of an inter-snapshot linear map, dynamic mode decomposition (DMD) of the pair-wise sequential snapshots was performed. The linear map **A** optimally describes the flow evolution over the small temporal separation $\Delta t$ and is given in matrix form as

$$\mathbf{X}_{\rm t+\Delta t} = \mathbf{A}\mathbf{X}_{\rm t} \qquad (1)$$

$$\mathbf{X}_{\rm t} \triangleq [\mathbf{v}'^1_{\rm t} \ \ldots \ \mathbf{v}'^N_{\rm t}]; \ \ \mathbf{X}_{\rm t+\Delta t} \triangleq [\mathbf{v}'^1_{\rm t+\Delta t} \ \ldots \ \mathbf{v}'^N_{\rm t+\Delta t}] \qquad (1a)$$

where $\mathbf{X}_{\rm t}$ and $\mathbf{X}_{\rm t+\Delta t}$ are the matrices that contain both the streamwise and transverse fluctuating components of the velocity fields obtained from the PIV images recorded from *Camera-1* and *Camera-2*, respectively. These are the column vectors $\mathbf{v}'_{\rm t}$ and $\mathbf{v}'_{\rm t+\Delta t}$, respectively, each with a dimension of $2(M_{\rm x} \times M_{\rm y})$, where $M_{\rm x}$ and $M_{\rm y}$ are the velocity

vector grid points in the axial and lateral directions, respectively. Here, *N* is the number of snapshots. The aforementioned formulation follows a similar approach introduced by Sikroria et al. [46,61]. The extraction of the dynamic characteristics of the linear process described by **A** is carried out by employing the *Exact DMD* algorithm of Tu et al. [39], which is outlined below. For brevity, $\mathbf{X}_t$ and $\mathbf{X}_{t+\Delta t}$ are denoted as **X** and $\mathbf{X}^{\#}$, respectively, so that Equation (1) is re-written as,

$$\mathbf{X}^{\#} = \mathbf{AX} \qquad (2)$$

**Algorithm:**

1.  Compute the singular value decomposition (SVD) of **X**

$$\mathbf{X} = \mathbf{U\Sigma V}^{\mathrm{T}} \qquad (3)$$

2.  Truncate the SVD

$$\mathbf{X}_{\mathrm{r}} = \mathbf{U}_{\mathrm{r}}\mathbf{\Sigma}_{\mathrm{r}}\mathbf{V}_{\mathrm{r}}^{\mathrm{T}} \qquad (4)$$

3.  Define the matrix **Ã**

$$\tilde{\mathbf{A}} = \mathbf{U}_{\mathrm{r}}^{\mathrm{T}}\mathbf{A}\mathbf{U}_{\mathrm{r}} = \mathbf{U}_{\mathrm{r}}^{\mathrm{T}}\mathbf{X}^{\#}\mathbf{V}_{\mathrm{r}}\mathbf{\Sigma}_{\mathrm{r}}^{-1} \qquad (5)$$

4.  Perform eigenvalue decomposition of $\tilde{\mathbf{A}}$ ($\mu_{\mathrm{j}}$ are the DMD eigenvalues)

$$\tilde{\mathbf{A}}\tilde{v}_{\mathrm{j}} = \mu_{\mathrm{j}}\tilde{v}_{\mathrm{j}} \qquad (6)$$

5.  Compute the eigenvalues of **A**

$$\lambda_{\mathrm{j}} = \frac{log(\mu_{\mathrm{j}})}{\Delta t} \qquad (7)$$

6.  Compute the DMD modes

$$v_{\mathrm{i}} = \frac{1}{\mu_{\mathrm{i}}}\mathbf{X}^{\#}\mathbf{V}_{\mathrm{r}}\mathbf{\Sigma}_{\mathrm{r}}^{-1}\tilde{v}_{\mathrm{i}} \qquad (8)$$

7.  Compute the corresponding DMD modes amplitudes

$$\alpha_{\mathrm{i}} = \mu_{\mathrm{i}}\frac{1}{\tilde{v}_{\mathrm{i}}}\mathbf{\Sigma}_{\mathrm{r}}\mathbf{V}_{\mathrm{r}}^{\mathrm{T}} \qquad (9)$$

The column vectors of the left singular matrix **U**, Equation (3), contain the POD modes of the data matrix **X**. Instead of performing a computationally intensive projection of the linear mapping matrix **A** onto all the *N* POD modes, a low-rank approximation of its singular value decomposition (SVD) is realised using only the first significant POD modes based on their specific turbulent kinetic energies, Equation (4). The matrix defined as $\tilde{\mathbf{A}}$, Equation (5), is a correlation between the matrix of energetically dominant POD modes $\mathbf{U}_{\mathrm{r}}$ and that of those shifted over the temporal separation $\mathbf{AU}_{\mathrm{r}}$. The DMD modes, Equation (8), are the spatial fields that represent the coherent structures in the flow, and their corresponding eigenvalues, Equation (7), provide information about their temporal evolution, i.e., growth/decay rates and the corresponding frequencies. The amplitude of the DMD modes is then given by Equation (9), which is a slightly modified form of the DMD amplitude matrix derived in [62].

The critical parameters for DMD are the temporal separation $\Delta t$ between the pairwise sequential velocity field snapshots and the number of snapshots *N*. An explanation of the selected value of $\Delta t$ has already been provided in Section 2.2. As DMD involves, at first, the computation of the POD modes, and then the projection of **A** onto these modes, an assessment of the number of snapshots required to approximate the dominant features of the jet dynamics was carried out. The results of this assessment and the threshold for truncation of the SVD are presented in the next section.

## 4. Results

### 4.1. Jet Characterisation

The initial flow conditions of the jet at an axial location 0.17*d* downstream of the nozzle exit, which is taken to be the location at which *x*/*d* = 0, are shown in Figure 4a. The profiles

are that of the mean axial velocity $\overline{U}$ and the axial component of the normal Reynolds stress $\overline{u'^2}$, along the radial coordinate $r$. These mean statistics are that of the 10,000 statistically independent velocity samples obtained from the recordings of *Camera-1* and are also the average of the values on either side of the jet center-line $y = 0$ ($r = 0$). The mean velocity and Reynolds stress are normalised by the local maximum mean axial velocity $\overline{U}_{\mathrm{max}}$ and its square, respectively, whereas the radial coordinate is normalised by the half width $r_{1/2}$, which is defined as the radial location where the local mean axial velocity becomes half its maximum value ($r$ at $\overline{U}/\overline{U}_{\mathrm{max}} = 0.5$). Only every fourth data point is shown in Figure 4a for clarity. As the results in Figure 4a show, the jet exit velocity has a top-hat profile with the turbulence intensity (square root of the streamwise Reynolds stress) in the shear layer attaining a maximum value of $\approx$10% of $\overline{U}_{\mathrm{max}}$ at $r = r_{1/2}$.

**(a)**

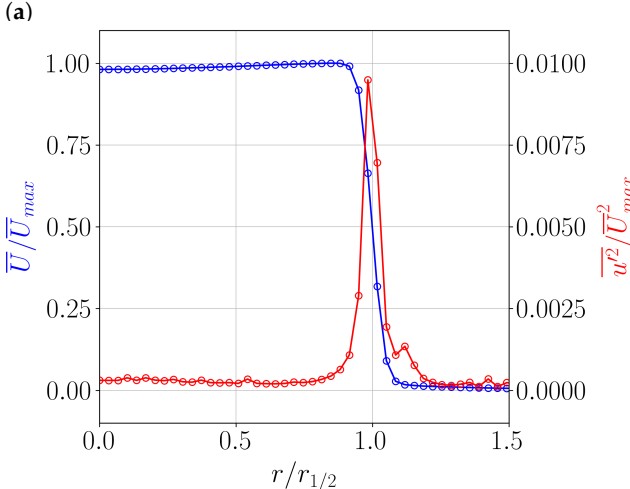

**(b)**

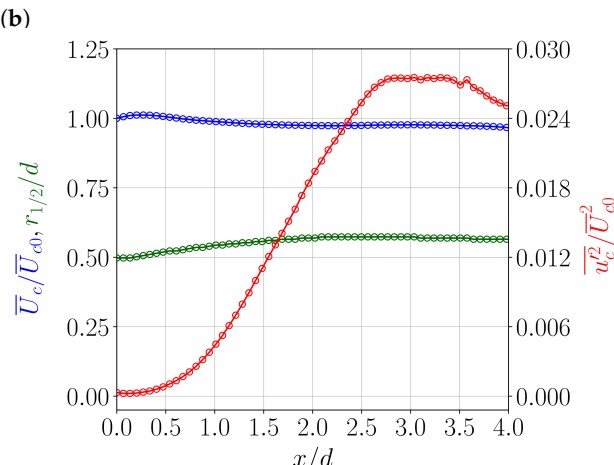

**Figure 4.** Jet flow characterisation: (**a**) mean axial velocity $\overline{U}/\overline{U}_{\mathrm{max}}$ and axial component of the streamwise Reynolds stress $\overline{u'^2}/\overline{U}_{\mathrm{max}}^2$ at the jet exit; (**b**) axial variation of the jet centre-line mean axial velocity $\overline{U}_{\mathrm{c}}/\overline{U}_{c0}$, centre-line axial component of the streamwise Reynolds stress $\overline{u_{\mathrm{c}}'^2}/\overline{U}_{c0}^2$, and the half width $r_{1/2}/d$.

The jet spread and the variation of the mean jet characteristics along the centre-line are shown in Figure 4b. Only every fourth data point is shown in this figure for clarity. The half width $r_{1/2}$ serves as a measure of the width and spread of the jet and is normalised by the jet diameter $d$ in this case. Its variation along the axial coordinate, which is also normalised by $d$, shows only a minor increase from the initial value of 0.5 at the jet exit. The centre-line

mean axial velocity $\overline{U}_c$ and the centre-line axial component of the streamwise Reynolds stress $\overline{u'^2_c}$ are normalised by the jet-exit mean centre-line axial velocity $\overline{U}_{c0}$ and its square, respectively. The normalised centre-line velocity increases very slightly as the jet exits the converging nozzle and then comparably decreases downstream, but stays very close to the value of 1.0, which indicates that the potential core of the jet spans the experimental domain. The centre-line streamwise Reynolds stress increases exponentially downstream until it attains a plateau between the locations $x/d = 2.75$ and 3.5, and where it reaches a value $\approx 2.7$ times that of the maximum value in the initial shear layer.

### 4.2. Assessment of Sample Size Sufficiency

The DMD formulation relies only on the recorded input data, in this case the pair-wise sequential velocity field snapshots, and does not need any information about the underlying linear mapping matrix $\mathbf{A}$. This feature enables processing only a smaller sub-domain of the entire measurement region where the relevant dynamics occur, which in this case is the near-field of the jet. The selected sub-domain spans $2d \times 1d$ ($x \times y$) from the jet exit and is highlighted by the yellow rectangle in Figure 5, which is an instantaneous snapshot of the jet flow obtained from *Camera-2*. The snapshot shows the velocity vectors overlaid on the contour of the axial velocity $u$, which is normalised by the instantaneous centre-line jet exit axial velocity $u_{c0}$. The DMD sub-domain includes the initial development of the jet shear layer, which typically is dominated by a linear instability mechanism, and that is followed by the formation of a periodic array of vortices via the Kelvin–Helmholtz instability mechanism. Approximately three streamwise wavelengths of the vortex structures are present within the axial extent of the DMD sub-domain.

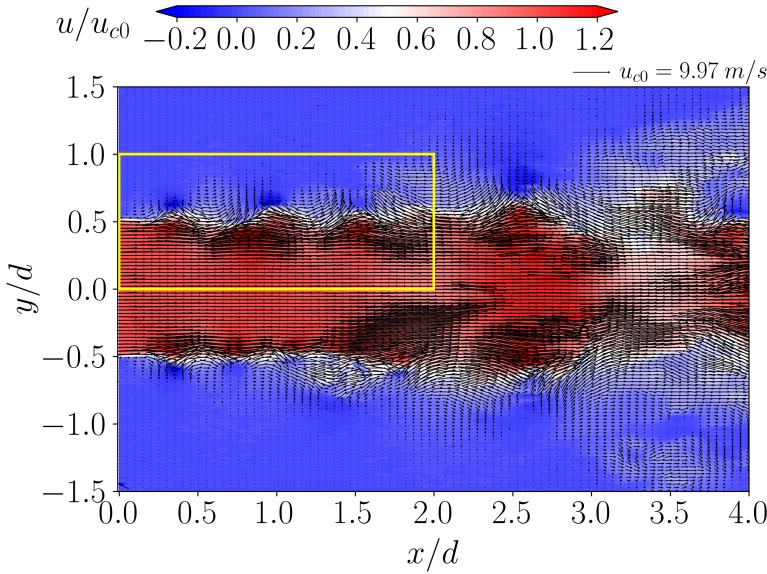

**Figure 5.** Instantaneous velocity vectors overlaid on the instantaneous axial velocity contour $u/u_{c0}$, with the DMD sub-domain highlighted by the yellow rectangle.

A suitable sample size for the modal decomposition is determined from the reference population that comprises the entire stored dataset of $N_{\text{ref}} = 10{,}000$ snapshots obtained from *Camera-1*. This involves the creation of the input data matrix $\mathbf{X}$ using the reference dataset and computation of its SVD as per Equation (3). The square of the singular values in the diagonal matrix $\mathbf{\Sigma}$ correspond to the turbulent kinetic energies of each of the $N_{\text{ref}}$ POD *Modes* and form the basis of ranking them in terms of the captured flow-field fluctuating energy. The energy contribution of each of the first 50 POD *Modes* as a fraction of the total turbulent kinetic energy (sum of the squares of the singular values) and their cumulative energies are shown in Figure 6. The energy fraction of *Mode* 10 and that of the following modes is less than 2% of the total energy, and the cumulative energy of the first 20 *Modes*

captures more than 85% of the total energy. The latter quantity is used as the metric to determine the sample size required to extract energetically important features of the jet flow in the chosen sub-domain.

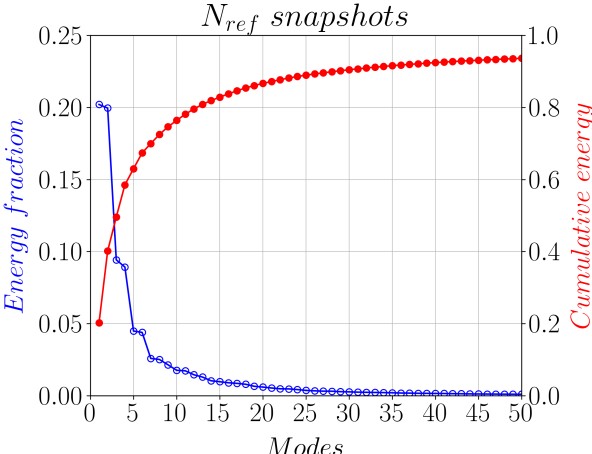

**Figure 6.** Fraction of the total kinetic energy and cumulative kinetic energies of the POD modes of $\mathbf{X}_t$ dataset with $N_{\text{ref}}$ = 10,000 snapshots.

A sample size sufficiency test was undertaken through a random selection of $N$ unique snapshots from the reference dataset of $N_{\text{ref}}$ snapshots, followed by the computation of the POD of that sub-sample dataset. This trial of random sample selection and its POD computation was performed 200 times to calculate the statistical estimator, which in this case is the 95% confidence interval of the cumulative energy of the first 20 *Modes* for a given sub-sample dataset $(CE_{20}^N)_{0.95}$. Five sub-sample datasets were examined with $N = [500, 1000, 2000, 3000, 4000]$ snapshots. The variation of the statistical estimator with the sub-sample dataset sizes is shown in Figure 7, wherein the statistical estimator is expressed as the percentage of the cumulative energy of the first 20 POD *Modes* of the reference dataset $CE_{20}^{N_{\text{ref}}}$. $(CE_{20}^N)_{0.95}$ decreases as the sub-sample size increases, which although expected, occurs at a faster rate between $N = 500$ and $2000$ snapshots compared with that between $N = 2000$ and $4000$ snapshots. Hence, an intermediate sub-sample size of $N = 3000$ snapshots in the latter interval was chosen to be sufficient, for which $(CE_{20}^N)_{0.95}$ is only around 0.2% that of $CE_{20}^{N_{\text{ref}}}$.

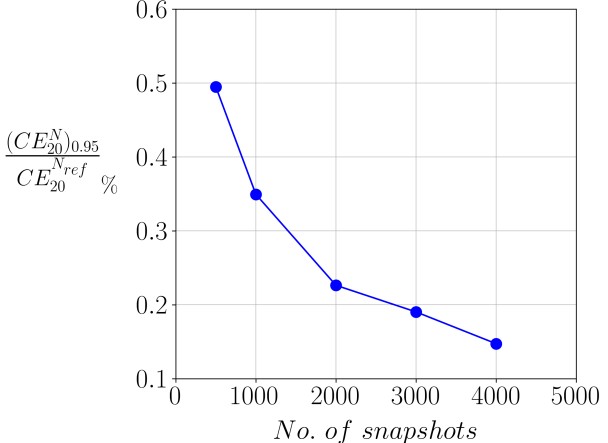

**Figure 7.** Variation of the 95% confidence interval of the cumulative energy of the first 20 POD *Modes* $(CE_{20}^N)_{0.95}$ with the examined sub-sample dataset sizes, expressed as the percentage of the cumulative energy of the first 20 POD *Modes* of the reference dataset $CE_{20}^{N_{\text{ref}}}$.

With the adequate number of snapshots determined from the aforementioned analysis, the input data matrix $\mathbf{X}_t$ (or $\mathbf{X}$) is then created as per Equation (1a) with $N = 3000$. However, not all those $N$ POD *Modes* are used in the next steps of the DMD algorithm. The SVD of $\mathbf{X}$ is truncated to include only the first 9 POD *Modes* onto which $\mathbf{A}$ is projected, because the energy contribution of each of the remaining modes to the total energy is very small (see Figure 6).

### 4.3. DMD Results

Since the dynamics of the extracted dominant POD modes are described by $\tilde{\mathbf{A}}$, and with both $\mathbf{X}_r$ and $\mathbf{X}^{\#}$ being used to compute it (Equation 5), a statistical approach for *Exact DMD* was adopted. This involved a random but pair-wise sequential selection of $N = 3000$ snapshots from the reference datasets of $N_{\text{ref}}$ snapshots obtained from *Camera-1* and *Camera-2*, to create $\mathbf{X}$ and $\mathbf{X}^{\#}$ matrices, respectively, and followed by the computation of the DMD algorithm (Equations (3)–(9)). This trial of random selection of snapshot pairs for the creation of the input data matrices and computation of their DMD was performed 200 times to calculate the statistics of the DMD eigenvalues, modes and amplitudes, the results of which are presented in this section.

The DMD mean eigenvalues are displayed in the complex plane in Figure 8, with all the eigenvalues lying in the stable (bottom) half plane $\lambda_{\Re} < 0$. The eigenvalues have been normalised by the jet integral-time scale $d/U_j$. As input matrices consist of real-valued data, the eigenvalue spectrum is symmetric with respect to the imaginary axis $\lambda_{\Im} = 0$. The eigenvalue with a zero imaginary value corresponds to a non-oscillatory DMD mode that decays, and is thus not important for the dynamic analysis.

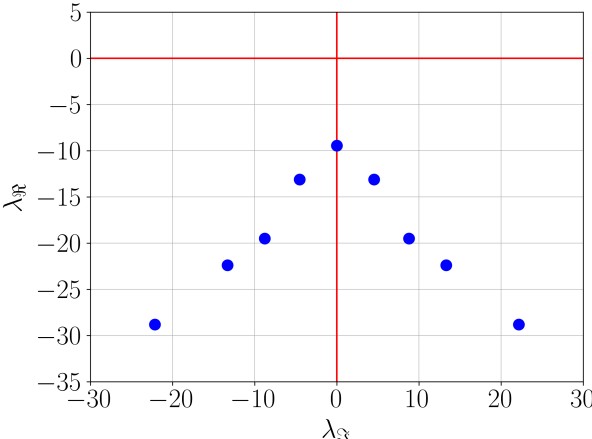

**Figure 8.** Spectrum of the DMD mean eigenvalues $\lambda$ in the complex plane.

The amplitude spectrum presented in Figure 9 shows the mean amplitudes of the DMD modes against their corresponding mean Strouhal numbers. The amplitudes shown are the sum of the squares of the real and imaginary components of each DMD amplitude computed from Equation (9), and have been scaled by the maximum amplitude of all the modes. The Strouhal number of the DMD mode is calculated as $St = \lambda_{\Im}\theta/2\pi U_j$, where $\theta$ and $U_j$ are as defined in Section 2.2. The amplitudes occur in pairs for each DMD mode that has a non-zero imaginary eigenvalue component, hence there are only five data points in the spectrum. The vertical and horizontal error bars on the data points are that of the 95% confidence intervals of the corresponding quantities, whereas the vertical broken lines are of $St_n = 0.032$ and its four higher harmonics. The computed non-zero $St$ values of the DMD modes very closely match those of $St_n$ and three of its higher harmonics, with the largest amplitude being that of the DMD mode at the second harmonic ($2St_n$).

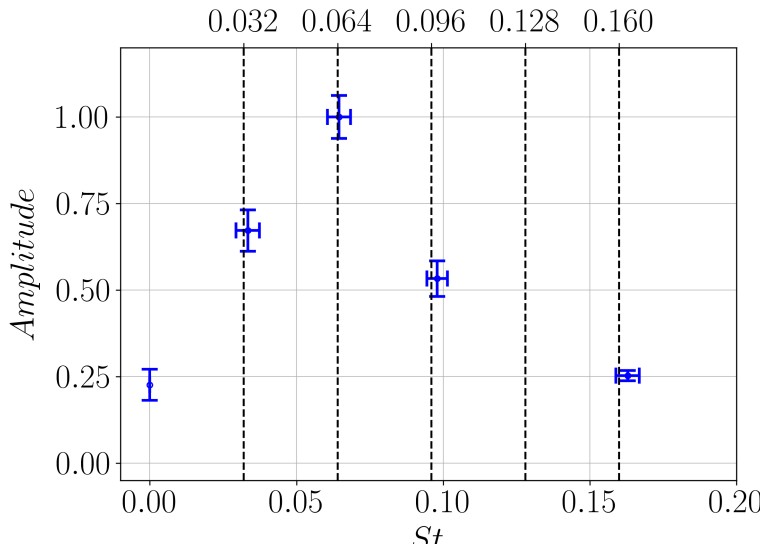

**Figure 9.** Spectrum of the DMD mean amplitudes and Strouhal numbers (*St*). The vertical and horizontal error bars on the data points are that of the 95% confidence intervals of the amplitude and *St*, respectively. The vertical broken lines are of $St_n = 0.032$ and its harmonics $2St_n$, $3St_n$, $4St_n$ and $5St_n$.

The DMD mean modes spanning the shear layer ($y/d \leq 1$) in the near-field region of the jet ($x/d \leq 1$) are presented in Figure 10. These are shown in the form of the fluctuating velocity vectors (left column) and contours of the streamwise and transverse fluctuating velocity components (middle and right columns, respectively). Only the real parts of the DMD mean modes are shown here, which have been normalised by their corresponding maximum quantities. The imaginary part (not shown) is phase-shifted by $\pi/2$. The fundamental coherent dynamic mode ($St = 0.033$) is characterised by large-scale vortical structures with approximately one oscillatory period over the streamwise extent of the displayed near-field region. The second harmonic mode ($St = 0.064$) has similar spatial features but with one and a half streamwise oscillatory periods. This spatial scaling increases to one oscillatory period in the third harmonic mode ($St = 0.098$) and then reduces to approximately two oscillatory periods in the fifth harmonic mode ($St = 0.163$). Not unexpectedly, the coherent structures develop more small-scale spatial features with increasing frequency (increasing *St*).

The aforementioned observations entail an analysis of the streamwise variation of the energy content of the DMD modes. The energy of each DMD mode is given by its $L^2$-norm of the complex streamwise and transverse components of the DMD velocity modes which is integrated along the cross-stream or transverse direction. The streamwise variation of this energy on a log scale, normalised by the energy at the jet exit ($x/d = 0$), is shown in Figure 11a. The energies of the fundamental and second harmonic DMD modes are higher than those of the other two harmonic modes in the region of $x/d \geq 0.5$, and where the energies of the latter two modes begin to decay. As the linear dynamics of the jet shear layer occur during its initial development, the focus is on the region very close to the jet exit ($0 \leq x/d \leq 0.25$) where the energies grow exponentially, which is representative of linear dynamics. This region is highlighted by the cyan coloured rectangle in Figure 11a.

The gradient of the DMD mode energy along the streamwise direction exemplifies the downstream spatial growth rate (amplification) of the modes. The streamwise variation of the amplification in the highlighted near-field region, normalised by the amplification at the jet exit, is shown in Figure 11b. The amplification of all the DMD modes is linear in the region $0.03 \leq x/d \leq 0.18$, with the amplification of the second harmonic DMD mode approximately equal to that of the fundamental mode. The same equality applies for the third and fifth harmonic modes; however, the amplification of these two modes is lower.

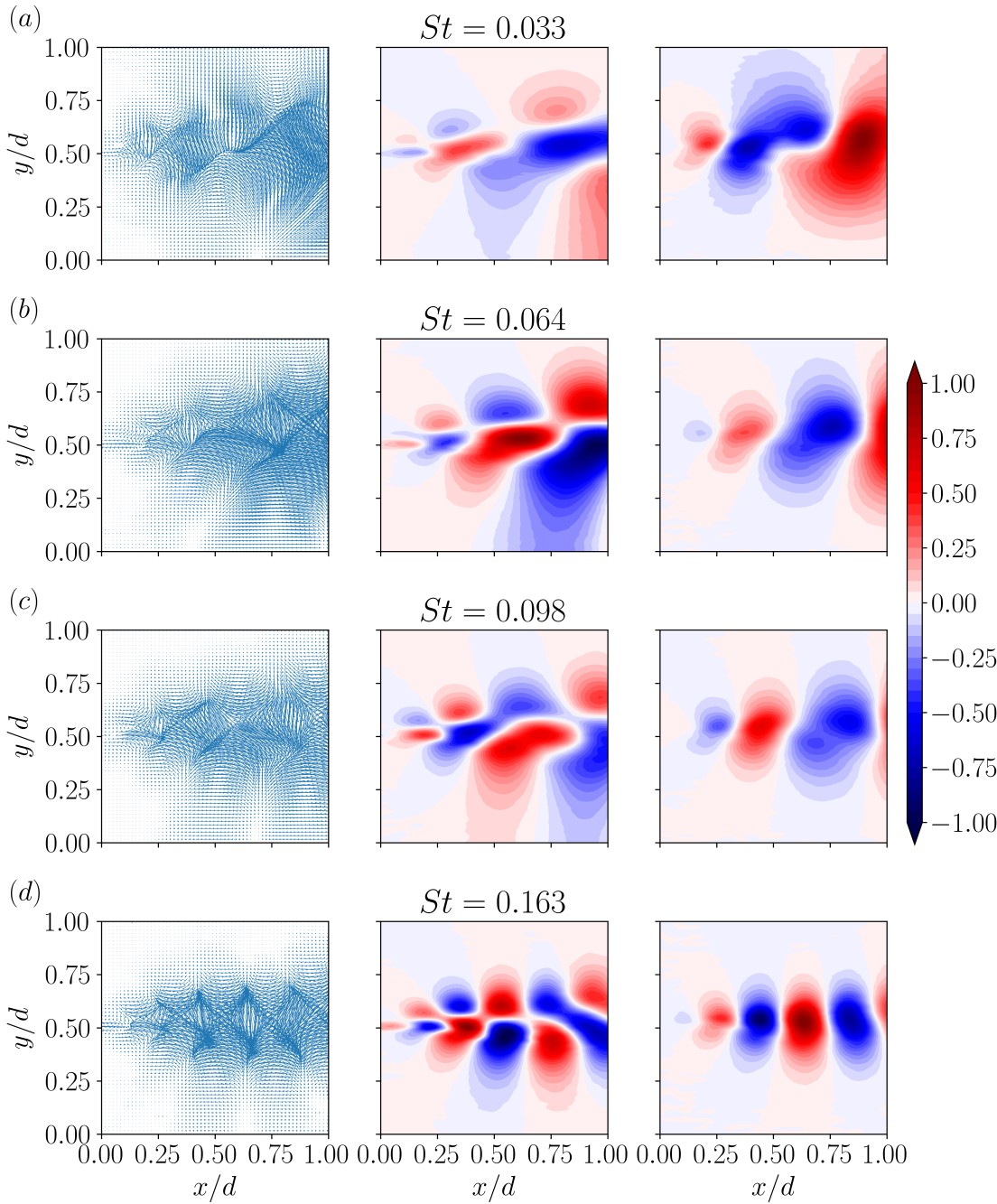

**Figure 10.** DMD modes visualised by the real part of the fluctuating velocity vectors (left column) and contours of their streamwise (middle column) and transverse (right column) components corresponding to (**a**) $St = 0.033$; (**b**) $St = 0.064$; (**c**) $St = 0.098$; (**d**) $St = 0.163$.

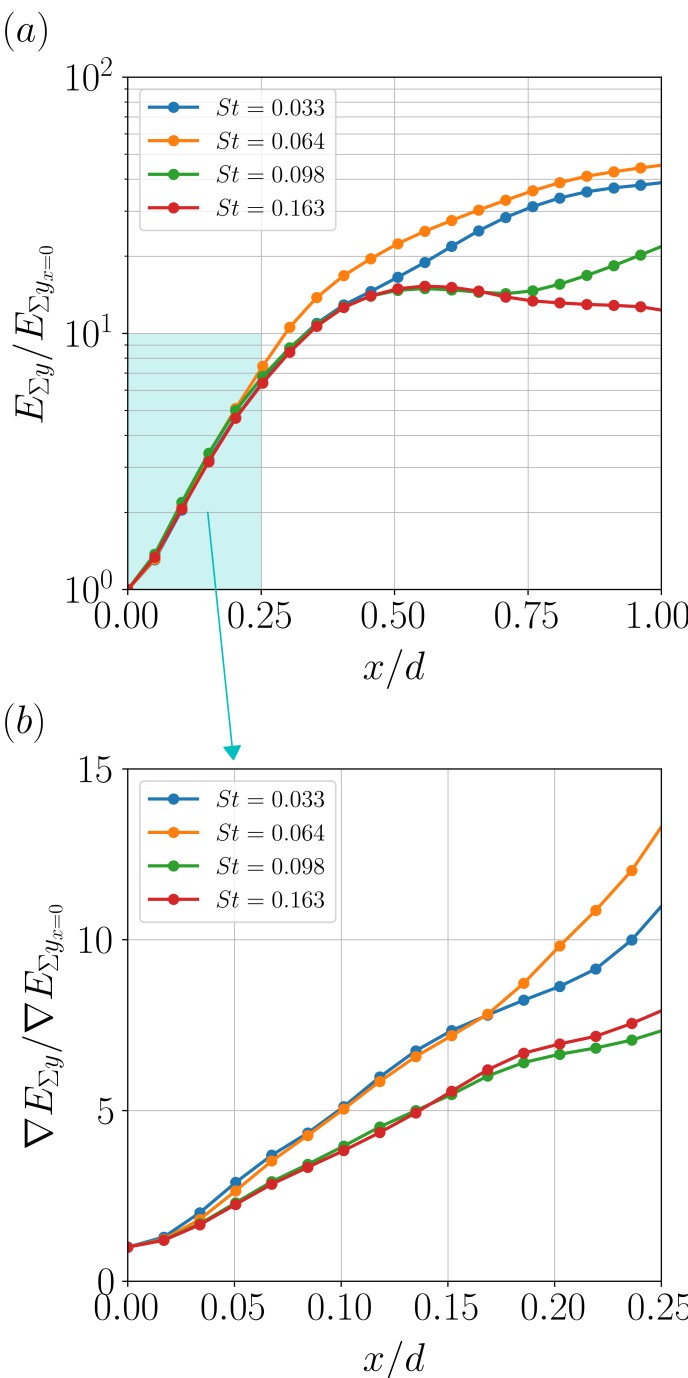

**Figure 11.** Streamwise variation of the DMD mode energies: (**a**) energies in the near-field region of the jet; (**b**) gradient of the energies in the cyan highlighted region.

## 5. Discussion

A temporal analysis using high-spatial resolution *Dual-PIV* coupled with *Exact DMD* has enabled the extraction of coherent structures that accurately describe the flow dynamics. These coherent structures, although dynamically relevant, may not necessarily capture the dominant energy content of the flow. However, in the present case, the knowledge of the modal energies (Figure 6) has been used to discriminate signal from noise and to enable the identification of the dynamic modes using the first nine energetically significant POD *Modes*. The extraction of the fundamental DMD mode corresponding to $St = 0.033$, which matches the $St_n$ of the most spatially amplified wave as predicted by classical linear stability theory [63], demonstrates that the chosen low-dimensional linear system captures

the temporal dynamics of the near-field region of the turbulent jet flow. This is noteworthy because this result is a consequence of data, which are not time-resolved fluid velocity time series, but rather a set of statistically independent pair-wise time-resolved velocity vector fields.

The fundamental DMD mode ($\lambda = (-13.1002, \pm 4.5387)$ in Figure 8) corresponds to the most dominant structure of all the coherent dynamic modes in the near-field shear layer of a high-speed free-jet, which is also the most persistent as it decays the slowest, with the higher harmonic modes decaying progressively faster. In contrast, the amplitude of the second harmonic mode is the highest, followed by that of the fundamental mode (Figure 9). This latter observation conforms with the streamwise energy variation of the corresponding DMD modes (Figure 11a). The gradient of the modal energies shows that spatial amplification of the DMD modes in the linear regime can be grouped in two pairs where each has two modes with the same amplification (Figure 11b). One of these pairs is formed by the fundamental and the second harmonic mode, indicating that both these modes are the most spatially amplified in the near-field jet region, whereas the result from the linear stability theory suggests that this occurs only for the fundamental mode $St_n$. The reason for this can potentially be attributed to the jet velocity profile in the present study (Figure 4a), which is different from the model *tanh* velocity profile used in the study by Michalke and Hermann [63]. Nevertheless, the streamwise variation of the amplification of all the DMD modes is linear in the near-field jet region, which is similar to the streamwise variation of the maxima of the amplitude distribution in the jet shear layer that was observed in the experiments by Freymuth [64].

## 6. Conclusions

High-spatial resolution *Dual-PIV* coupled with *Exact DMD* was used in this study to extract and analyse the linear dynamics of a turbulent round high-speed free-jet. The *Dual-PIV* measurements provided pair-wise time-resolved velocity field snapshots for the *Exact DMD* analysis in a sub-domain of the near-field of the jet. The computed modal energies with a statistical approach were used to determine the adequate number of velocity snapshots required for the *Exact DMD* analysis.

The resulting DMD modes show high spatially resolved coherent structures in the jet shear layer, and the spectra specifies their frequencies, which comprises a fundamental frequency with three higher harmonics. The fundamental DMD mode correlates with the most spatially amplified instability wave as predicted by classical linear stability theory. However, the second harmonic mode, which has the highest amplitude among all the modes, is also found to be the most spatially amplified in the near-field jet shear layer.

Overall, the experimental and data decomposition technique presented in this paper demonstrates a novel and powerful methodology to determine the underlying flow dynamics in high-speed turbulent shear flows using non-sequential temporally resolved pairs of velocity flow fields with high-spatial resolution.

**Author Contributions:** Conceptualization, J.S.; methodology, J.S. and T.S.; experiments, A.D.; formal analysis, V.C.; investigation, A.D. and V.C.; data curation, A.D. and V.C.; writing—original draft preparation, V.C.; writing—review and editing, V.C., A.D., T.S., J.S. and C.A.; supervision, J.S. All authors have read and agreed to the published version of the manuscript.

**Funding:** The research was supported by the Australian Research Council. The research also benefited from high-performance computing (HPC) resources provided through the National Computational Merit Allocation Scheme (NCMAS). These resources were provided via the facilities of the National Computational Infrastructure (NCI), the Pawsey Supercomputing Centre, and the Multi-modal Australian ScienceS Imaging and Visualisation Environment (MASSIVE) at Monash University. Both NCMAS and the participating facilities are funded by the Australian Government.

**Data Availability Statement:** The data presented in this study are available on request from the corresponding author.

**Conflicts of Interest:** The authors declare no conflict of interest.

## Abbreviations

The following abbreviations are used in this manuscript:

| | |
|---|---|
| DMD | Dynamic Mode Decomposition |
| PIV | Particle Image Velocimetry |
| SVD | Singular Value Decomposition |

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
