# Peer review of "Investigating the Linear Dynamics of the Near-Field of a Turbulent High-Speed Jet Using Dual-Particle Image Velocimetry (PIV) and Dynamic Mode Decomposition (DMD)"

_fluids, doi:10.3390/fluids8020073_

Round 1

Reviewer 1 Report

Very nice paper and work aimed to overcome current hardware limitations.

Author Response

The authors thank the reviewer for their positive comments.

Reviewer 2 Report

Taken as a whole, this is an interesting analysis of the Linear Dynamics of the Near Field of a Turbulent High-speed Jet by use of Dual-PIV and DMD. In fact, DMD method is very suitable for Dual PIV. However, there are still a few places that could be improved. 

1. Since this paper is an experimental study of the linear dynamics of the near field of a turbulent high-speed jet, I hope that the authors can briefly quote the existing theoretical or numerical simulation related research results in the introduction, especially for the frequency of oscillation and its corresponding temporal growth/decay rate.

2. In section 4.3 (DMD results), I wonder if the fundamental DMD mean amplitudes (Figure 9) or the streamwise variation of the fundamental DMD modes energies (Figure 11(b)) can be compared with the linear stability theory result. If the comparison is too difficult, please explain briefly to me.

3.Because two-component -two-dimensional (2C–2D) PIV measurements is adopted in this paper, I want to know whether and how to confirm the axial symmetry of the jet.

Author Response

Please see the attachment which contains a point-by-point response to the reviewer's comments and the revised manuscript with the changes highlighted.

Reviewer 3 Report

GENERAL COMMENTS

This study suggests a novel approach that combines dual particle image velocimetry with dynamic mode decomposition to accomplish extraction of time-resolved velocity fields with a high spatial-resolution. The manuscript is well-written and structured, and of broad scientific interest. However, the novelty of the manuscript is not properly highlighted. Also, some details necessary for the understanding of the procedures followed by the authors are missing. Finally, the typesetting and definition of the mathematical symbols needs polishing.

The authors need to close the discussion with some ideas for future research, maybe how the suggested approach can be applied beyond jet streams. For example, PIV is very useful in hydrological applications, especially of extreme events, where videos of streamflows (usually citizen-shot footage) is sometimes the only available data source. However, recent studies have indicated that traditional PIV tools can significantly underestimate the stream velocities in the cases where the flow velocity is high and non-uniform (e.g., locations of hydraulic jump) (Rozos et al., 2022). The suggested approach could provide a more robust (extract coherent flow structures) and accurate interpretation of this valuable source of information.

SPECIFIC COMMENTS

Location: section 2.3

Comment: What platform/tool is used for PIV algorithms and for DMD?

Location: lines 76 -78

Comment: Is this combination or the application to this specific kind of problems the basic novelty?  In the previous lines, there are references to 4 studies for Dual-PIV, some of them 20-year-old, and to one study, almost 10-year-old, for Exact DMD. The novelty of this study needs to be documented more explicitly. What is new in this study compared to previous applications of Dual-PIV and Exact DMD? 

Location: Figure 1

Comment: The "Perspec cage" is not mentioned in the previous lines. Is this the "enclosed square test-section of side 42d and height 80d"?

Location: line 123

Comment: "The laser, shutter and the twin-cameras ... " -> "The laser, the shutter and the twin-cameras ..."

Location: Figure 3

Comment: The durations of Pulse A and Pulse B are different. Why?

Location: Equations (1) - (9)

Comment 1: Please use consistent typesetting for mathematical symbols (doi: 10.13140/RG.2.2.10775.21922).

Comment 2: Textual subscripts or superscripts are not italic (e.g., "^T" for transpose, "_r" for truncated).

Comment 3: Equation (6), if u_j is a vector (eigenvector) it should be both bold and italic.

Comment 4: Equation (1a), N needs to be defined here, not in the last lines of this section.

Comment 5: Equation (1a), what are the dimensions of v_t, v_(t+Dt)

Comment 6: Equation (9), is this a vector (the eigenvector of Equation (6)) in the denominator?

Comment 7: Equation (5), in general, in SVD decomposition, the dimensions of X are (m x n), the dimensions of U are (m x m), the dimensions of Sigma are (m x n), and of V (n x n). If so, how can this equation be dimensionally consistent?

REFERENCES

Rozos, E.; Mazi, K.; Lykoudis, S. On the Accuracy of Particle Image Velocimetry with Citizen Videos—Five Typical Case Studies. Hydrology 2022, 9, 72. https://doi.org/10.3390/hydrology9050072

Author Response

(The authors gave the same response as above.)
